# NRG1 is a critical regulator of differentiation in TP63-driven squamous cell carcinoma

Ganapati V Hegde[1†]*, Cecile de la Cruz[2], Jennifer M Giltnane[3], Lisa Crocker[2], Avinashnarayan Venkatanarayan[1], Gabriele Schaefer[2], Debra Dunlap[3], Joerg D Hoeck[4], Robert Piskol[5], Florian Gnad[5], Zora Modrusan[6], Frederic J de Sauvage[4], Christian W Siebel[1], Erica L Jackson[1‡]*

[1]Discovery Oncology, Genentech, South San Francisco, United States; [2]Translational Oncology, Genentech, South San Francisco, United States; [3]Pathology, Genentech, South San Francisco, United States; [4]Molecular Oncology, Genentech, South San Francisco, United States; [5]Bioinformatics, Genentech, South San Francisco, United States; [6]Molecular Biology, Genentech, South San Francisco, United States

**Abstract** Squamous cell carcinomas (SCCs) account for the majority of cancer mortalities. Although TP63 is an established lineage-survival oncogene in SCCs, therapeutic strategies have not been developed to target TP63 or it's downstream effectors. In this study we demonstrate that TP63 directly regulates NRG1 expression in human SCC cell lines and that NRG1 is a critical component of the TP63 transcriptional program. Notably, we show that squamous tumors are dependent NRG1 signaling in vivo, in both genetically engineered mouse models and human xenograft models, and demonstrate that inhibition of NRG1 induces keratinization and terminal squamous differentiation of tumor cells, blocking proliferation and inhibiting tumor growth. Together, our findings identify a lineage-specific function of NRG1 in SCCs of diverse anatomic origin.
DOI: https://doi.org/10.7554/eLife.46551.001

*For correspondence:
gvhegde@gmail.com (GVH);
ericajackso@gmail.com (ELJ)

Present address: †ORIC
Pharmaceuticals, South San
Francisco, United States;
‡Pharmacyclics, Sunnyvale,
United States

Competing interest: See
page 11

Reviewing editor: William C
Hahn, Dana-Farber Cancer
Institue, United States

## Introduction

Within the past decade, lineage addiction has emerged as a common paradigm to explain how certain tumors depend on co-opted survival and self-renewal programs that drive the normal development of the tissues from which they arise (*Garraway and Sellers, 2006*). During normal development and tissue homeostasis, 'master regulator' transcription factors control large sets of genes regulating cellular identity, differentiation and survival (*Chan and Kyba, 2013*). Amplifications of master regulators act as oncogenic drivers in cancers arising in the tissues whose development they normally control. Examples include MITF, which directs melanocyte development and is amplified in some melanomas and NKX2.1, which directs development of the distal lung epithelium and is amplified in some lung adenocarcinomas (*Kendall et al., 2007*). Some cancers remain fully dependent on transcription factors expressed by precursor cells of the lineage from which they develop, even in the absence of genetic alterations in these genes. Examples include AR in prostate cancer and ESR1 in luminal breast cancers (*Garraway and Sellers, 2006*).

Despite varied anatomic origins, squamous cell cancers (SCCs) share many common properties, including genetic and epigenetic alterations (*Dotto and Rustgi, 2016*). The TP63 transcription factor exemplifies an important lineage dependency in SCCs (*Ramsey et al., 2013*). Amplification of *TP63* is prevalent in SCCs, and TP63 expression is used to distinguish SCCs from other cancer subtypes in multiple tissues (*Dotto and Rustgi, 2016*). SCCs arise in numerous organ systems that contain

stratified or pseudo-stratified epithelia, including the lung, head and neck, esophagus, skin, bladder and cervix. Interestingly, like other lineage survival oncogenes, TP63 is a key regulator of the progenitor cells in the basal cell compartment during normal development and homeostasis of most stratified or pseudostratified epithelia (*Mills et al., 1999*; *Yang et al., 1999*). Despite its established role as a driver of lineage dependency, TP63 is a transcription factor, and as such is challenging to target therapeutically.

Here we show that NRG1 expression is directly regulated by TP63 in SCCs of various organs, and that co-expression of NRG1 and its receptor ERBB3 is prevalent in SCCs. Moreover, we find that many of the SCC models that co-express NRG1 and ERBB3 depend on NRG1 autocrine signaling in vivo, in contrast to non-squamous cancers that exhibit NRG1 autocrine signaling but are not dependent on it.

## Results and discussion

### TP63 regulates NRG1 expression

TP63 is highly expressed in basal cells of various epithelia and is required for the progenitor cell function. In addition, TP63 acts as a key survival factor and driver of SCCs (*Rocco et al., 2006*; *Thurfjell et al., 2005*). Interestingly, studies of normal mammary basal cells established that TP63 can directly activate NRG1 transcription (*Forster et al., 2014*). Therefore, we evaluated whether this transcriptional wiring exists in SCCs. We found that *NRG1* and *TP63* expression significantly correlated in both esophageal and lung squamous cell carcinomas (LUSC) as determined from The Cancer Genome Atlas (TCGA) transcriptome data (*Figure 1A*). TP63 has two isoform classes that either contain (TA-TP63) or lack (DeltaN-TP63) an N-terminal transactivation domain. Despite lacking this domain, deltaN-TP63, the major isoform expressed in SCCs, functions as both a positive and negative transcriptional regulator of different target gene subsets (*Hibi et al., 2000*; *Moll and Slade, 2004*). We evaluated whether TP63 regulates NRG1 expression in SCCs using siRNAs to knockdown all *TP63* isoforms (siTP63 # 14) or only the *TA-TP63* isoforms (siTA-TP63 # 13) and assessing expression of both isoforms of the NRG1 EGF-like domain, *NRG1α* and *NRG1β*, by qPCR in the OE-21 and KYSE-140 SCC cell lines. Knockdown of just the TA isoforms modestly reduced *NRG1* expression, whereas silencing of all isoforms robustly decreased *NRG1* expression (*Figure 1B*). We further expanded this finding in an additional cell line, KYSE-180 (*Figure 1C*). Knockdown of *deltaN-TP63* significantly reduced *NRG1* transcripts, confirming that deltaN-TP63 regulates NRG1 expression in SCCs. Immunoblotting confirmed knockdown of deltaN-TP63 at the protein level. In addition, we determined whether NRG1 is a direct transcriptional target of deltaN-TP63 by ChIP-PCR using antibodies for TP63alpha and deltaN-TP63 and PCR primers that amplify the NRG1 promoter. Binding of both TP63 isoforms was significantly enriched at the region −30 kB from the transcriptional start site (TSS), which encompasses the TP63 binding motif, compared to the control locus at −21 kB (*Figure 1D*). Together, these data suggest that deltaN-TP63 directly regulates NRG1 transcription in SCC.

### Efficacy of anti-NRG1 in in vivo models of SCC

Emerging evidence suggest that high ERBB3 or high NRG1 expression is associated with poor clinical outcome in SCCs (*Qian et al., 2015*). We reasoned that in order for NRG1 to promote SCC growth, tumors would have to co-express NRG1 and its receptor ERBB3. Interestingly, *NRG1/ERBB3* co-expression appeared prevalent in lung and head and neck SCCs, but was notably rare in lung adenocarcinoma across the various cancer datasets available in TCGA (*Figure 2A*). To ascertain whether SCCs co-expressing NRG1 and ERBB3 are responsive to inhibition of NRG1 signaling, we screened a panel of cell lines from SCC indications including lung, esophageal and skin for growth sensitivity to an NRG1 blocking antibody (*Hegde et al., 2013*) in vitro. Anti-NRG1 treatment modestly inhibited the growth of cell lines expressing both NRG1 and ERBB3 (*Figure 2B*). Because the p63 transcriptional program is critical for both maintenance and cell fate determination of epithelial progenitor cells, we reasoned that perturbation of this program would be more impactful in vivo. We assessed the effect of NRG1 inhibition on xenograft tumors derived from the FaDu head and neck, HCC95 lung and KYSE-180 esophageal squamous cell carcinoma lines. Importantly, anti-NRG1 treatment markedly inhibited tumor growth in each of these models to an extent that far exceeded

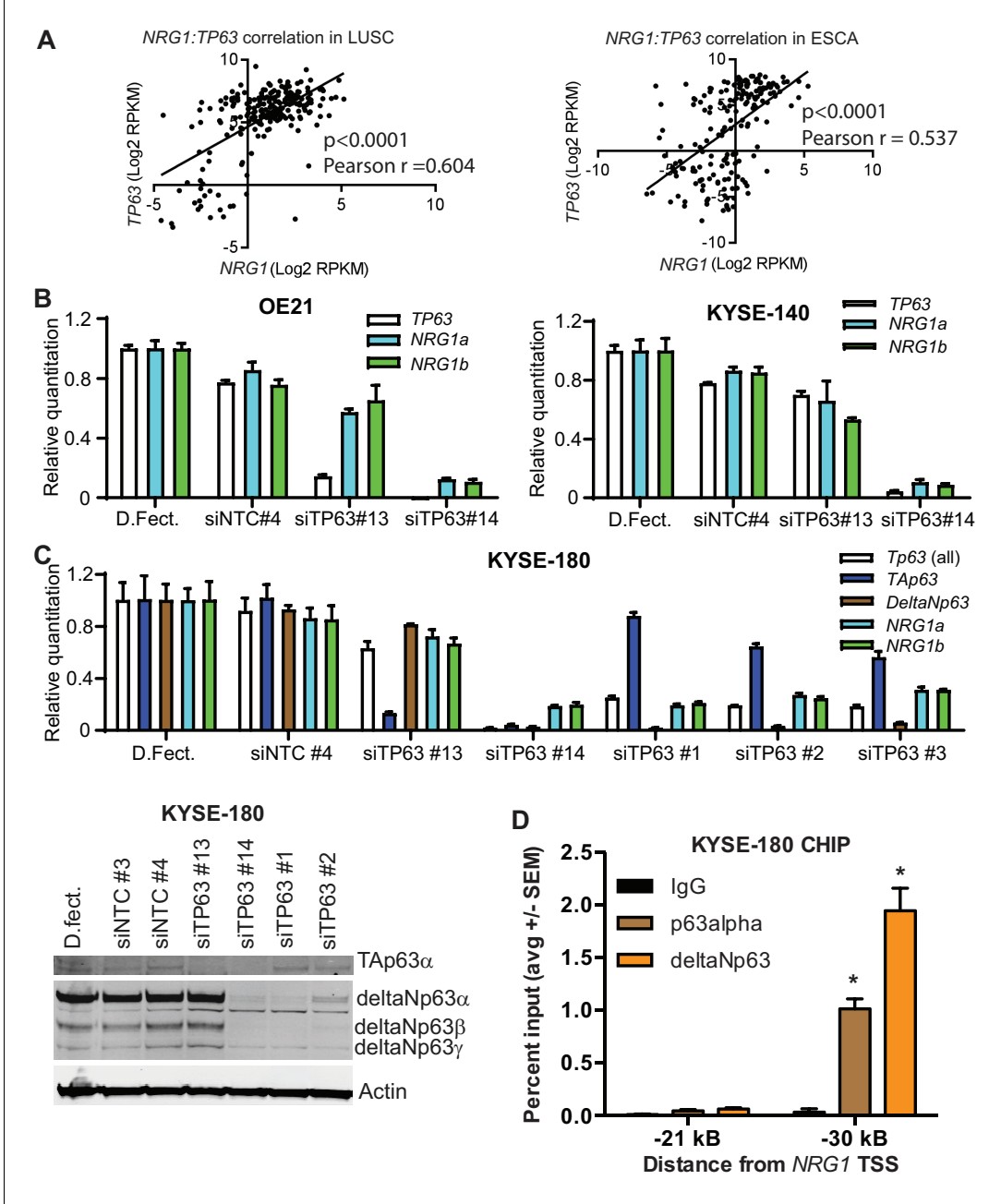

**Figure 1.** TP63 regulates NRG1 in SCC. (**A**) Correlation of *TP63* and *NRG1* in LUSC (n = 223) and Esophageal (n = 263) patient samples in TCGA data set. Statistical significance was determined by two-tailed test and Pearson correlation (r) was determined. (**B**) Relative expression of *TP63*, *NRG1α* and *NRG1β* upon *TP63* knockdown using siRNA in OE21 and KYSE-140 SCCs. (**C**) Relative expression of *TP63* (all, TA and deltaN isoforms), *NRG1α*, *NRG1β* upon *TP63* knockdown using siRNA in KYSE-180 SCC. Expression of TA and deltaN *TP63* and upon *TP63* knockdown by siRNA. siRNAs for TA-TP63 (#13), all isoform-TP63 (#14), deltaN-TP63 [#1, #2, #3] and non-target control (NTC#3, NTC#4). Relative quantitation by qPCR in B and C is mean with standard deviation relative to dharmafect transfection reagent control. (**D**) ChIP analysis of p63alpha and deltaNp63 binding −30 kB from the NRG1 transcriptional start site (TSS) in KYSE-180 SCC. IgG and primers amplifying the −21 kB region were used as controls. Data is represented as average with standard error of mean from three independent experiments. One-way ANOVA test was used to determine the statistical significance in comparison to IgG. *p<0.05.

DOI: https://doi.org/10.7554/eLife.46551.002

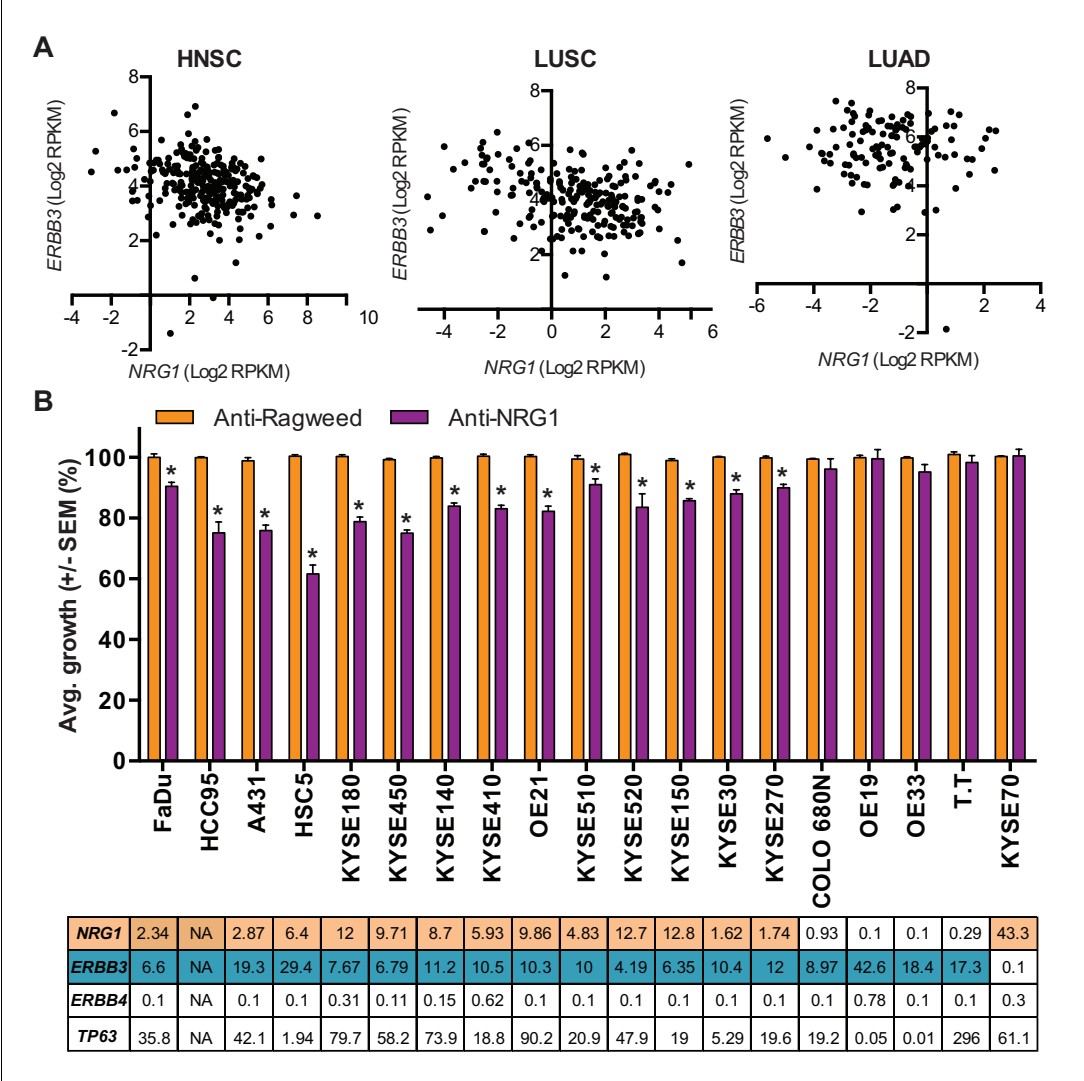

| | FaDu | HCC95 | A431 | HSC5 | KYSE180 | KYSE450 | KYSE140 | KYSE410 | OE21 | KYSE510 | KYSE520 | KYSE150 | KYSE30 | KYSE270 | COLO 680N | OE19 | OE33 | T.T | KYSE70 |
|---|---|---|---|---|---|---|---|---|---|---|---|---|---|---|---|---|---|---|---|
| **NRG1** | 2.34 | NA | 2.87 | 6.4 | 12 | 9.71 | 8.7 | 5.93 | 9.86 | 4.83 | 12.7 | 12.8 | 1.62 | 1.74 | 0.93 | 0.1 | 0.1 | 0.29 | 43.3 |
| **ERBB3** | 6.6 | NA | 19.3 | 29.4 | 7.67 | 6.79 | 11.2 | 10.5 | 10.3 | 10 | 4.19 | 6.35 | 10.4 | 12 | 8.97 | 42.6 | 18.4 | 17.3 | 0.1 |
| **ERBB4** | 0.1 | NA | 0.1 | 0.1 | 0.31 | 0.11 | 0.15 | 0.62 | 0.1 | 0.1 | 0.1 | 0.1 | 0.1 | 0.1 | 0.1 | 0.78 | 0.1 | 0.1 | 0.3 |
| **TP63** | 35.8 | NA | 42.1 | 1.94 | 79.7 | 58.2 | 73.9 | 18.8 | 90.2 | 20.9 | 47.9 | 19 | 5.29 | 19.6 | 19.2 | 0.05 | 0.01 | 296 | 61.1 |

**Figure 2.** Modest but significant growth inhibition of SCCs by anti-NRG1 treatment in vitro. (**A**) NRG1 and ERBB3 mRNA levels in head and neck squamous carcinoma (HNSC), lung squamous carcinoma (LUSC) and lung adenocarcinoma (LUAD) patient samples from TCGA. (**B**) In vitro growth of different SCC cell lines in the presence of anti-NRG1 or control anti-Ragweed antibody. Expression levels of *NRG1*, *Erbb3*, *Erbb4* and *TP63* (RPKM) for each cell line are shown below the respective in vitro growth data. Average growth is presented as is mean with standard error of mean relative to anti-Ragweed from four independent experiments with more than three replicates in every experiment. Statistical significance was determined using t-test with * indicates p<0.05.

DOI: https://doi.org/10.7554/eLife.46551.003

that observed in vitro. (*Figure 3A*). In addition, increased keratinization and changes in tumor cell morphology were observed in tumors from anti-NRG1 treated mice (*Figure 4—figure supplement 1*). To further evaluate the NRG1-dependency in ERBB3/NRG1 co-expressing squamous cell cancers, we tested the efficacy of anti-NRG1 in lung SCC PDX models. Again, anti-NRG1 significantly inhibited the tumor growth of three models that co-express NRG1 and ERBB3, resulting in tumor stasis (*Figure 3B* and *Figure 3—figure supplement 1*).

Although NRG1 is not widely recognized as an important factor in cutaneous SCC, downregulation of endogenous NRG1 expression occurs during differentiation of cultured keratinocytes, and activation of NRG1 signaling inhibits keratinocyte differentiation and promotes neo-epidermal outgrowth in human skin explant cultures (*De Potter et al., 2001*). Furthermore, mice with Krt5-Cre-driven knockout of *Erbb3* in basal progenitor cells are resistant to carcinogen-induced skin tumorigenesis and exhibit defective wound healing (*Dahlhoff et al., 2015*), while *Krt5*-driven

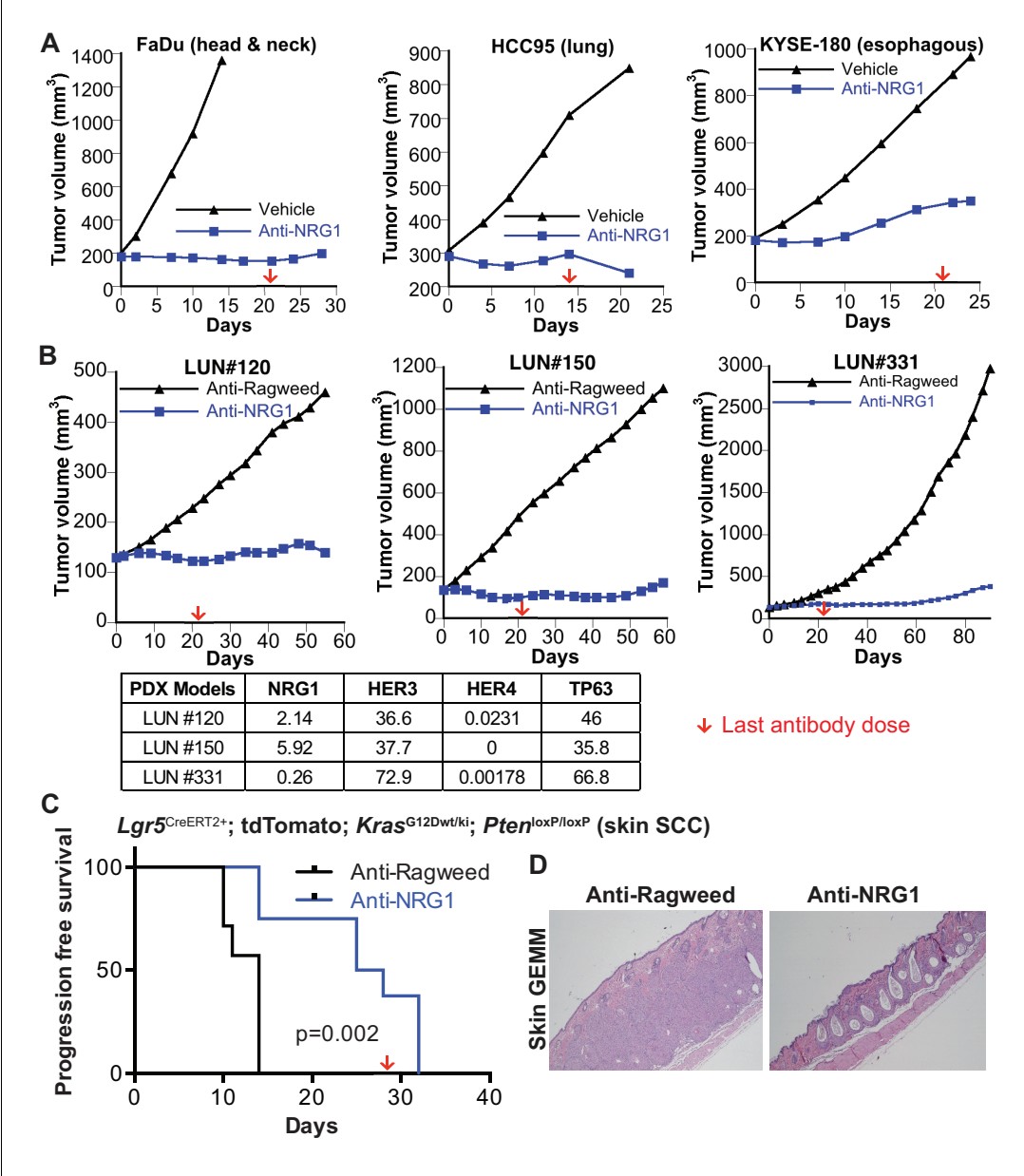

↓ Last antibody dose

| PDX Models | NRG1 | HER3 | HER4 | TP63 |
|---|---|---|---|---|
| LUN #120 | 2.14 | 36.6 | 0.0231 | 46 |
| LUN #150 | 5.92 | 37.7 | 0 | 35.8 |
| LUN #331 | 0.26 | 72.9 | 0.00178 | 66.8 |

**Figure 3.** Anti-NRG1 significantly inhibits the tumor growth in vivo of SCCs that express both NRG1 and ERBB3. Effect of anti-NRG1 or anti-Ragweed (control) antibodies in (**A**) FaDu (head and neck), HCC95 (lung) and KYSE-180 (esophagus) SCC cell lines in preclinical mouse xenograft models, N = 8–12 mice/group, (**B**) Lung SCC PDX models, n = 4 mice/group, and (**C**) Lgr5$^{CreERT2}$; Pten$^{flox/flox}$; Kras$^{LSL-G12D/+}$ skin GEMM, n = 7–8 mice/group. Statistical significance was determined by Log-rank test. Red arrow indicates the time of the final antibody dose. (**D**) H and E staining for GEMM skin at the end of study.

DOI: https://doi.org/10.7554/eLife.46551.004

The following figure supplements are available for figure 3:

**Figure supplement 1.** Anti-NRG1 inhibits in vivo tumor growth in SCC models.
DOI: https://doi.org/10.7554/eLife.46551.005

**Figure supplement 2.** Anti-NRG1 does not inhibit the growth of ovarian models expressing NRG1 and ERBB3 receptor in vivo.
DOI: https://doi.org/10.7554/eLife.46551.006

overexpression of deltaN-TP63 enhances carcinogen-induced skin tumorigenesis (*Devos et al., 2017*). Therefore, we tested the effect of inhibiting NRG1 in the $Lgr5^{CreERT2}$; $Pten^{flox/flox}$; $Kras^{LSL-G12D/+}$ genetically engineered model of cutaneous SCC. In this highly aggressive tumor model, anti-NRG1 dramatically increased the progression free survival (PFS) compared to control treatment, nearly doubling time to progression from 14 to 27 days (p<0.002) (*Figure 3C–D*). Progression was defined as enlargement and redness of the lips and snout, as this was the only clinical observation in the animals and macroscopic skin tumors were observed only upon necropsy. In control animals, the dermis was expanded by confluent nests of squamous cells forming tumors that > 80% of the dermis in skin lesions sampled from 5 of 7 animals. In contrast, in 5 of 6 anti-NRG1 treated animals, lesions were limited to central dilated keratinization without dermal expansion, consistent with epidermal inclusion cyst, a benign squamous proliferation (*Figure 3D*).

Our analysis of RNAseq data across tumor indications also revealed prevalent *NRG1/ERBB3* co-expression in ovarian cancer. We tested the effect of anti-NRG1 on a panel of ovarian cancer cell lines and indeed proliferation of cell lines expressing both NRG1 and ERBB3 was inhibited by anti-NRG1 in vitro (*Figure 3—figure supplement 2A*). However, unlike the SCC models, ovarian PDX models expressing NRG1 and ERBB3 were not sensitive to anti-NRG1 treatment in vivo (*Figure 3—figure supplement 2B–C*) suggesting that dependence on NRG1 signaling may be specific to the TP63 driven cancer types. Together these data support a role for NRG1 in mediating p63 lineage dependency in SCCs.

## Anti-NRG1 induces squamous differentiation

To explore the mechanism mediating the robust tumor growth inhibition observed in anti-NRG1 treated SCC models, we repeated the in vivo studies for three SCC xenograft models of different anatomic origins and collected tumors after treatment. Histological analysis revealed that in all models anti-NRG1-treated tumors exhibited a more well-differentiated appearance, with increased eosinophilic cytoplasm consistent with enhanced keratinization, indicating that anti-NRG1 treatment was driving differentiation in these tumors (*Figure 4—figure supplement 1* and *Figure 4A*). Moreover, anti-NRG1-treated tumors showed a dramatic increase in the expression of KRT10, a differentiation-specific keratin normally restricted to the post-mitotic layers of stratified-keratinizing and cornifying epithelia (*Figure 4A*). Immunoblot analyses of treated tumors showed that anti-NRG1 inhibited ERBB3 activation and decreased the levels of multiple proliferation markers, consistent with a mechanism in which differentiation is induced at the expense of proliferation (*Figure 4B* and *Figure 4—figure supplement 2*). Markers of apoptosis were not affected (*Figure 4B*).

To broadly assess changes in expression of differentiation markers, we analyzed RNAseq data, focusing on a gene expression signature of human airway basal cells (*Hackett et al., 2011*). Anti-NRG1 treatment caused significant changes in the levels of nearly all the genes in the panel (*Figure 4C*). Notably, many of the genes upregulated by anti-NRG1 treatment are associated with differentiation of stratified epithelia, such as *KLK7*, *BNC1* and *ADAMTS1*, demonstrating a profound effect on the differentiation state of the tumor cells that becomes more pronounced with ongoing treatment. Of note, NRG1 is itself a marker of airway basal cells, and anti-NRG1 treatment resulted in downregulation of the NRG1 transcript. Upon differentiation, airway basal cells cluster with keratinocytes in unsupervised analyses (*Hackett et al., 2011*). Therefore, we evaluated markers of keratinocyte differentiation and found that anti-NRG1 treatment resulted in strong upregulation of several well-established keratinocyte differentiation markers (*Figure 4D* and *Figure 4—figure supplement 2*), while the expression of several lung progenitor cell markers was decreased (*Figure 4D* and *Figure 4—figure supplement 2*). Treatment of SCC cell lines with anti-NRG1 in vitro also increased expression of the differentiation markers *KRT1*, *KRT10*, *IVL* and *KRTDAP* (*Figure 4—figure supplement 3*). However, the magnitude of upregulation was much lower than in the tumors, consistent with the more modest effects of anti-NRG1 treatment on the proliferation of cell lines in vitro.

NRG1 signaling supports progenitor cell function and regeneration in a diverse set of normal tissues (*Bartus et al., 2016*; *Bersell et al., 2009*). Our data suggest that NRG1 may serve a similar function downstream of TP63 in tumors derived from squamous epithelia. We show that treating squamous tumors with an NRG1 blocking antibody reduced proliferation, concomitant with induced cellular differentiation and increased keratin production. The marked difference in response between in vitro and in vivo models is consistent with a pro-differentiation effect of anti-NRG1 treatment, as cancer cells are known to undergo dedifferentiation and lose tissue-specific expression patterns and

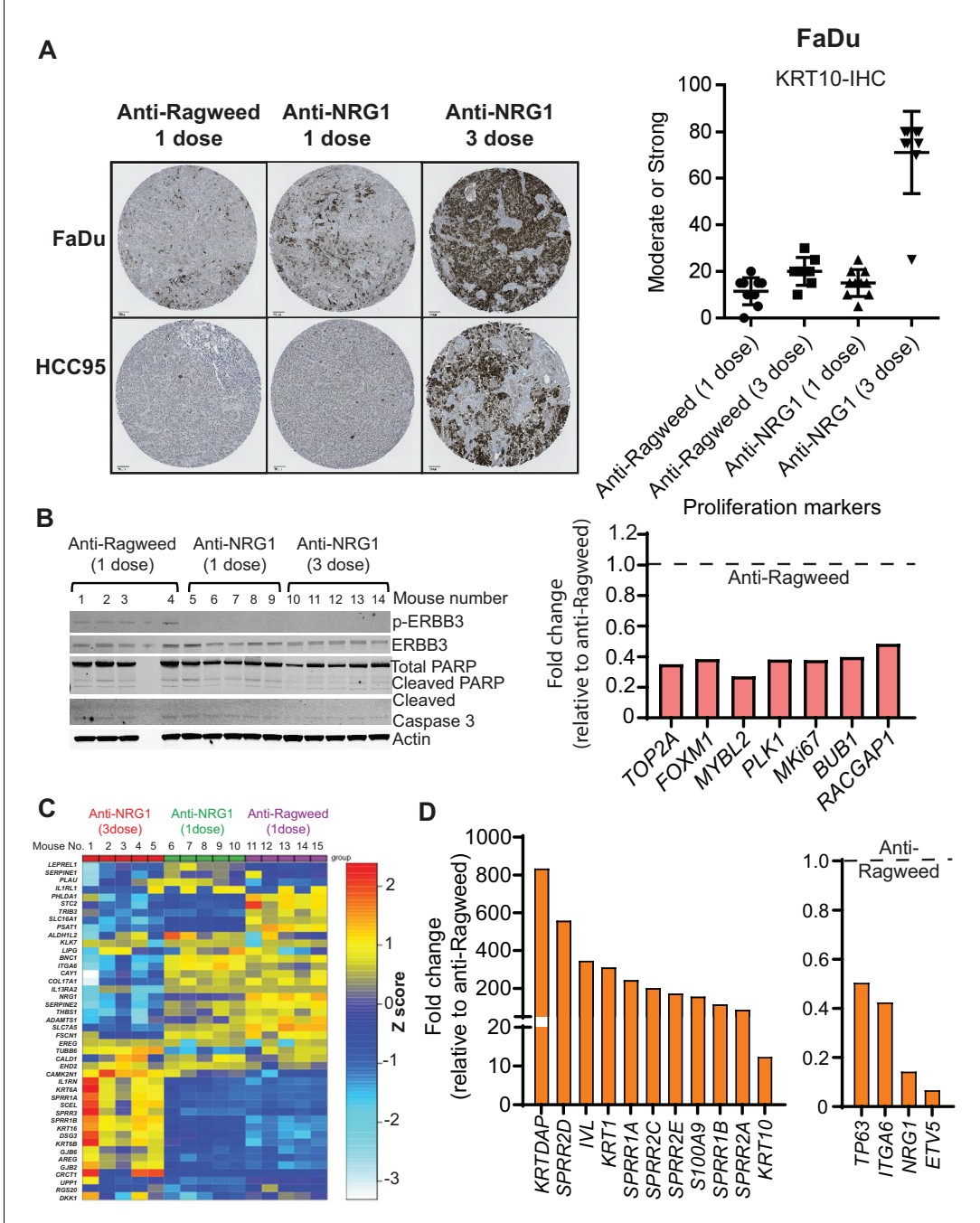

**Figure 4.** Anti-NRG1 induces squamous differentiation and inhibits proliferation in SCCs. (**A**) Representative image of KRT10 (differentiation marker) staining of tumor from FaDu and HCC95 SCC models upon anti-NRG1 treatments compared to anti-Ragweed control. N = 5 mice/group. (**B**) Protein levels of apoptosis markers upon anti-NRG1 treatment in vivo in HCC95 SCC xenografts. phospho-ERBB3 level was used to assess inhibition of signaling. Expression of proliferation markers upon one and three doses of anti-NRG1 relative to anti-Ragweed treatment in vivo in HCC95 lung SCC by RNAseq. (**C**) Expression of lung basal cell differentiation markers after one or three doses of anti-NRG1 and one dose of anti-Ragweed treatment in HCC95 lung SCC xenograft tumors by RNAseq. N = 5 mice/group. Expression of (**D**) squamous differentiation markers and progenitor cell related markers following three doses of anti-NRG1 relative to anti-Ragweed treatment in HCC95 lung SCC xenograft tumors by RNAseq. Average fold change relative to anti-ragweed from n = 5 mice/group.

DOI: https://doi.org/10.7554/eLife.46551.007

The following source data and figure supplements are available for figure 4:

**Source data 1.** Panel of human airway basal cell gene signature in *Figure 4C* (as published by *Hackett et al., 2011*).

DOI: https://doi.org/10.7554/eLife.46551.011

*Figure 4 continued on next page*

*Figure 4 continued*

**Figure supplement 1.** Anti-NRG1 induces differentiation in SCC.
DOI: https://doi.org/10.7554/eLife.46551.008
**Figure supplement 2.** Anti-NRG1 inhibits proliferation and induces differentiation in SCC.
DOI: https://doi.org/10.7554/eLife.46551.009
**Figure supplement 3.** Anti-NRG1 increases differentiation markers in vitro in SCC.
DOI: https://doi.org/10.7554/eLife.46551.010

differentiation capacity when cultured long term on plastic. In contrast, although NRG1 and ERBB3 co-expression is prevalent in ovarian tumors, ovarian cancer models did not respond to NRG1 inhibition in vivo. Our model predicts such a result, given that fewer than 10% of epithelial ovarian tumors express p63 and it is not a lineage oncogene in this cancer type.

EGFR signaling is known to be an important driver in SCCs and anti-EGFR therapies have been approved for the treatment of head and neck and lung SCCs (*Sacco and Worden, 2016*; *Thakur and Wozniak, 2017*). In contrast, NRG1-ERBB3 signaling has been implicated as a resistance mechanism to anti-EGFR therapies (*Wheeler et al., 2008*). However, dual inhibition of ERBB3 and EGFR did not show significant clinical benefit compared to EGFR inhibition alone. Our findings on the role of NRG1 in SCCs raise the possibility that NRG1 may still provide resistance to EGFR therapeutics through ERBB4 receptor activation when ERBB3 is inhibited, and provide rationale for exploring the clinical benefit of NRG1 inhibition in combination with EGFR inhibition in SCCs.

# Materials and methods

## Key resources table

| Reagent type (species)or resource | Designation | Source or reference | Identifiers | Additional information |
|---|---|---|---|---|
| Strain, strain background (M. musculus) | C.B-17 SCID beige mice | Charles River Labs | CB17.Cg-Prkdc$^{scid}$Lyst$^{bg-J}$/Crl | |
| Strain, strain background (M. musculus) | Athymic nude mice | Harlan Sprague Dawley (Livermore facility) | Hsd:Athymic Nude-*Foxn1*$^{nu}$ | |
| Genetic reagent (M. musculus) | *Lgr5*$^{CreERT2+}$; tdTomato; *KrasG12D*$^{wt/ki}$; *PTEN*$^{loxP/loxP}$ | *Lgr5*$^{CreERT2+}$; pubmed id: 21927002 tdTomato; pubmed id: PMC2840225 *KrasG12D*$^{wt/ki}$; pubmed id: 11751630 *PTEN*$^{loxP/loxP}$; pubmed id: 11691952 | | LAR, Genentech |
| Antibody | Anti-Ragweed | Genentech | 9652 | In vivo: 20 mg/kg, i.p.,qwk In vitro: 20 ug/ml |
| Antibody | Anti-NRG1 | Genentech | YW538.24.71 | In vivo: 20 mg/kg, i.p.,qwk In vitro: 20 ug/ml |
| Antibody | deltaNTP63 (Rabbit Polyclonal IgG) | Biolegend | RRID: AB_2256361 Cat. #: 619001 | WB (1:1000) |
| Antibody | Actin (Mouse IgG) | BD Bioscience | RRID: AB_2289199 Cat. #: 612656 | WB (1:5000) |
| Antibody | p-ERBB3 (Rabbit mAb) | Cell Signaling Technologies | RRID: AB_2099709 Cat. #: 4791 | WB (1:500) |

*Continued on next page*

*Continued*

| Reagent type (species) or resource | Designation | Source or reference | Identifiers | Additional information |
|---|---|---|---|---|
| Antibody | ERBB3 (Rabbit mAb) | Cell Signaling Technologies | RRID: AB_2721919 Cat. #: 12708 | WB (1:500) |
| Antibody | Rabbit PARP (Rabbit polyclonal) | Cell Signaling Technologies | RRID: AB_2160739 Cat. #: 9542 | WB (1:1000) |
| Antibody | Cleaved caspase-3 (Rabbit mAb) | Cell Signaling Technologies | Cat. #: 9664 | WB (1:1000) |
| Antibody | TP63 alpha (Rabbit mAb) | Cell Signaling Technologies | RRID: AB_2637091 Cat. #: 13109 | CHIP (1:100) WB (1:1000) |
| Antibody | KRT10 (Rabbit polyclonal) | Covance Biologicals | RRID: AB_291580 Cat. #: PRB-159P | IHC (1:1000) |
| Commercial assay or kit | *TP63*-all | Life Technologies | Hs00978340_m1 | qPCR |
| Commercial assay or kit | TA-*TP63*-TA | Life Technologies | Hs00186613_m1 | qPCR |
| Commercial assay or kit | deltaN-*TP63* | Life Technologies | Hs00978339_m1 | qPCR |
| Commercial assay or kit | *NRG1a* | Life Technologies | Hs01103794_m1 | qPCR |
| Commercial assay or kit | *NRG1b* | Life Technologies | Hs00247624_m1 | qPCR |
| Commercial assay or kit | *KRTDAP* | Life Technologies | Hs00415563_m1 | qPCR |
| Commercial assay or kit | *KRT1* | Life Technologies | Hs00196158_m1 | qPCR |
| Commercial assay or kit | *KRT10* | Life Technologies | Hs00166289_m1 | qPCR |
| Commercial assay or kit | *IVL* | Life Technologies | Hs0846307_s1 | qPCR |
| Other | TA-TP63 | Dharmacon | #13 = J-003330–13 | siRNA |
| Other | all isoform-TP63 | Dharmacon | #14 = J-003330–14 | siRNA |
| Other | NTC#3 | Dharmacon | D-001810–03 | siRNA |
| Other | NTC#4 | Dharmacon | D-001810–04 | siRNA |
| Software, algorithm | GraphPad Prism | graphpad.com | RRID:SCR_002798 | |

## Cell culture

Cancer cell lines were sourced, authenticated, tested for mycoplasma and maintained by the Genentech cell bank (gCELL) as described (*Yu et al., 2015*). Cell lines were cultured in either RPMI-1640 or Dulbecco's modified Eagle's medium (DMEM) growth medium supplemented with 10% of fetal bovine serum (FBS), 2 mmoL glutamine and penicillin/streptomycin. For growth assays, tumor cells were cultured in media containing 2% FBS in 96-well plates, and treated for 96 hr with 20 ug/ml of anti-NRG1 YW538.24.71 (previously described in *Hegde et al., 2013*) or anti-Ragweed (control) antibodies in triplicate, and assayed using cell titer blue (Promega). For effect of antibodies on differentiation in vitro, FaDu, HCC95 and KYSE-180 tumor cells were cultured with 20 ug/ml of antibodies for three days. RNA was analyzed and qPCR was performed using ABI TaqMan primer/probes as explained below. Detailed characterization of this anti-NRG1 (YW538.24.71) was previously described (*Hegde et al., 2013*). Statistical significance was determined by t-test from at least three independent experiments. Expression of *NRG1*, *ERBB3* and *ERBB4* was assessed by RNAseq.

## Animal studies

All animal studies were approved by the Institutional Animal Care and Use Committee (IACUC) at Genentech (LASAR numbers 10-2319A, 16–1304, 16-1304A, 16–0098, 16–1120, 16–1143 and 16–2005, 16–1120). For cell line xenograft studies, tumor cells were subcutaneously inoculated into C.B-17 SCID beige mice (FaDu) or athymic nude mice (HCC95 and KYSE-180). Mice were randomized into treatment groups when tumor volumes reached ~200mm3. Antibodies were dosed once per week intraperitoneally (i.p.) at 20 mg/kg for three doses, and tumor volumes and body weights were measured twice a week. For RNAseq, histology and western blot, HCC95 tumors were collected 72 hr after the first or third dose. For patient-derived tumor xenografts, tumor fragments were subcutaneously implanted into Balb/C nude mice, which were randomized into treatment groups (n = 5) when tumor volumes reached 150–250 mm3. Dosing and measurements were performed as noted above for 3–4 weeks.

## Genetically engineered mouse model

The Lgr5$^{CreERT2+}$; tdTomato; KrasG12D$^{wt/ki}$; PTEN$^{loxP/loxP}$ squamous skin cancer mouse model (manuscript in preparation), was used following Genentech IACUC guidelines. At 10–15 weeks of age, mice were dosed once with Tamoxifen (100 mg/kg, i.p.). Three days after a single tamoxifen dose, mice were divided into two groups with mice of similar proportion of age and sex, and dosed with anti-Ragweed or anti-NRG1 antibodies (20 mg/kg, i.p., once in a week).

## Tumor growth analysis

A mixed-modeling approach was used to analyze tumor volumes for xenograft and PDX studies as explained earlier (Hegde et al., 2013). Cubic regression splines were used to fit a nonlinear profile to the time courses of log2 tumor volume for each treatment group and tumor volumes were plotted as fitted tumor volumes (mm$^3$). Percent tumor growth inhibition (TGI) was calculated relative to the average tumor volume of control (anti-Ragweed) mice. Kaplan-Meier analysis was used to determine progression-free survival for the skin GEMM SCC mice. Log-rank analysis was used for statistical analysis to compare treatment groups.

## RNA interference

Cells were plated in medium without antibiotics and transfected with 5 nmol/L of siRNA using Dharmafect transfection reagent (Dharmacon). siRNAs for TA-TP63 (#13 = J-003330–13), all isoform-TP63 (#14 = J-003330–14), deltaN-TP63 [#1 (S = GGACAGCAGCAUUGAUCAAUU, AS = UUGAUCAAUGCUGCUGUCCUU), #2 (S = CUUCUUAAGUAGAUUCAUAUU, AS = UAUGAAUCUACUUAAGAAGUU, #3(S = GGGACUUGAGUUCUGUUAUUU, AS = AUAACAGAACUCAAGUCCCUU)], and negative control non target control (NTC#3, NTC#4) were purchased from Dharmacon. RNA was isolated after 72 hr using a RNeasy Plus kit (Qiagen). cDNA was prepared using the Advantage RT for PCR kit (Clontech), and qPCR was performed using ABI master mix, and the results were validated by three independent experiments using the following Taqman assays (Life Technologies): *TP63*-all (Hs00978340_m1), TA-*TP63*-TA (Hs00186613_m1), deltaN-*TP63* (Hs00978339_m1), *NRG1a* (Hs01103794_m1) and *NRG1b* (Hs00247624_m1).

## Immunoblotting

72 hr after siRNA transfection, cells were washed twice with PBS and lysed in RIPA buffer containing protease/phosphatase inhibitors at 4˚C. Protein concentrations were determined by BCA (Thermo Scientific). Proteins were separated by SDS-PAGE, transferred to nitrocellulose membrane and stained with the indicated antibodies. HCC95 tumors from the in vivo studies were processed similarly for analysis of pharmacodynamic and apoptotic markers. Antibodies: TP63 alpha (Cell Signaling Technologies, CST 13109), deltaNTP63 (Biolegend 619001), actin (BD Bioscience 612656), p-ERBB3 (CST 4791), ERBB3 (CST 12708), PARP (CST 9542) and Cleaved caspase-3 (CST 9664).

## Chromatin immunoprecipitation (ChIP)

ChIP was performed from KYSE-180 cells (Diagenode iDeal CHIP-seq kit for Transcription factors). Cultures of 25 million cells were fixed and cross-linked with formaldehyde (1.1%) for 15 min at room temperature and stopped with glycine. The cell pellet was re-suspended in ChIP lysis buffers and

sonicated using a Bioruptor sonicator (Diagenode) to produce chromatin fragments averaging 200–500 bp. Sheared chromatin was incubated overnight at 4°C with protein A-coated magnetic beads plus either anti-TP63 alpha (CST 13109), anti-deltaNTP63 (BioLegend 619001) or isotype control antibody. Beads were washed, DNA was eluted and crosslinks were reversed during an incubation overnight at 65°C. Samples were treated with iPure beads and DNA was purified. Quantitative PCR was performed using sybr green PCR master mix (Applied Biosystem) to assess enrichment at the NRG1 promoter. The results were validated by three independent experiments. The ChIP ratio was calculated as enrichment over noise, normalized to the input. Statistical significance was determined by one-way ANOVA. Primers used for CHIP PCR were −21 kB NRG1 promoter (5'-TTCAAAAGG-GAGTGCCAACTTTTCC-3', 5'-GGTGCCTCACCTTTCTTCTTCCTGTCC-3') and −30 KB NRG1 promoter (5'-GCCCCAAATTCTTTTGCCCCTTAT-3', 5'-TTGGTTGGCTTGCTGAAGCTGGTGT-3') from NRG1 transcript start site as described earlier (*Forster et al., 2014*).

## Immunohistochemistry

Tumors collected after one or three dose of antibodies were fixed in 10% neutral-buffered formalin overnight then transferred to 70% ethanol, processed and embedded into paraffin. Tumor sections were subjected to H and E and IHC using rabbit polyclonal anti-Cytokeratin10 (KRT10) antibody (Covance Biologicals, PRB-159P), incubated at a concentration of 1.0 ug/mL for 60 min at room temperature and binding was visualized using ABC-Peroxidase Elite followed by DAB chromogen and counter stained with Mayer's hematoxylin. KRT10 expression was reviewed manually by a translational pathologist (JMG) and scored for the percentage of cells demonstrating moderate to strong immunoreactivity, excluding areas of necrosis (0–100%).

## RNA-seq

RNA-seq libraries were prepared using TruSeq RNA Sample Preparation kit (Illumina, CA) and sequenced on Illumina HiSeq 2500 sequencers, yielding an average of 34 million single-end reads (50 bp) per sample. Reads were aligned to the human genome version NCBI GRCh37 using GSNAP. Expression counts per gene were obtained by counting the number of reads aligned concordantly within a pair and uniquely to each gene locus as defined by NCBI, Ensembl gene annotations, and RefSeq mRNA sequences. Differential gene expression analysis was performed using edgeR. Gene enrichment analysis was performed on the edgeR differential expression results using the Gsea Pre-ranked tool available through the Broad's GSEA application. DESeq was used to compute the variance stabilized expression values for plotting the expression heat maps.

## Statistical analysis

Graphical and statistical data were generated with Microsoft Excel or GraphPad Prism (GraphPad Software, La Jolla, CA, USA). Statistical significance of differences between the results was assessed using a standard 2-tailed t-test or one-way ANOVA using Prism. $p < 0.05$ was considered statistically significant.

## Acknowledgements

The authors would like to thank gCELL, Genentech LAR, the Genentech pCORE laboratories and computational biology group and Elizabeth Lu for their help in various aspects of this work.

## Additional information

### Competing interests

Ganapati V Hegde, Cecile de la Cruz, Jennifer M Giltnane, Lisa Crocker, Avinashnarayan Venkatanarayan, Gabriele Schaefer, Debra Dunlap, Joerg D Hoeck, Robert Piskol, Florian Gnad, Zora Modrusan, Frederic J de Sauvage, Christian W Siebel, Erica L Jackson: employee of Genentech Inc at the time of participation in this study.

## Funding
The authors declare that there was no funding for this work

## Author contributions
Ganapati V Hegde, Conceptualization, Data curation, Formal analysis, Validation, Investigation, Methodology, Writing—original draft, Writing—review and editing; Cecile de la Cruz, Formal analysis, Managed PDX in vivo studies; Jennifer M Giltnane, Formal analysis, Histology data analysis; Lisa Crocker, Methodology, Performed FaDu of in vivo study; Avinashnarayan Venkatanarayan, Formal analysis, Conducted experiments, Analyzed data during manuscript revision; Gabriele Schaefer, Resources, Methodology, Provided some of reagents and involved in the discussion; Debra Dunlap, Methodology, Performed IHC; Joerg D Hoeck, Methodology, Developed skin SCC mouse model; Robert Piskol, Florian Gnad, Software, Formal analysis, Bioinformatics data analysis; Zora Modrusan, Resources, Methodology, RNAseq; Frederic J de Sauvage, Christian W Siebel, Writing—review and editing, Involved in discussion, writing and editing of the manuscript; Erica L Jackson, Conceptualization, Formal analysis, Supervision, Writing—original draft, Writing—review and editing

## Author ORCIDs
Ganapati V Hegde https://orcid.org/0000-0001-6473-153X
Erica L Jackson https://orcid.org/0000-0002-7100-8021

## Ethics
Animal experimentation: All animal studies were approved by the Institutional Animal Care and Use Committee (IACUC) at Genentech. The animal studies included herein were performed based on approved IACUC protocol numbers (LASAR 10-2319A, 16-1304, 16-1304A, 16-0098, 16-1120, 16-1143 and 16-2005, 16-1120).

## Decision letter and Author response
Decision letter https://doi.org/10.7554/eLife.46551.014
Author response https://doi.org/10.7554/eLife.46551.015

# Additional files

## Supplementary files
• Transparent reporting form
DOI: https://doi.org/10.7554/eLife.46551.012

## Data availability
All data generated or analysed during this study are included in the manuscript and supporting files. Source data file has been provided for Figure 4.

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
