## [Decision Letter]

Thank you for submitting your article "NRG1 is a critical component of the TP63 transcriptional program driving squamous cell cancers in multiple tissues" for consideration by *eLife*. Your article has been reviewed by three peer reviewers, one of whom is a member of our Board of Reviewing Editors, and the evaluation has been overseen by Kevin Struhl as the Senior Editor. The reviewers have opted to remain anonymous.

The reviewers have discussed the reviews with one another and the Reviewing Editor has drafted this decision to help you prepare a revised submission.

Summary:

The reviewers believe that the findings are of potential significance and may have therapeutic implications. Although NRG1 has already been established as a TP63 target in breast cells and NRG1/ERBB signaling has already been reported to play a role in squamous cell carcinoma biology, a strength of the work is the use of human material, murine models and cancer cell lines as well as an antibody to block NRG, which suggests that blocking NRG may affect tumor maintenance. However, the reviewers request additional work to more clearly demonstrate the interaction of NRG and p63, to clarify the specificity of some of the reagents and to determine whether squamous differentiation is the basis of the observed findings.

Essential revisions:

1) The title is quite broad. The reviewers were unclear whether the findings are true in all squamous tissues or are specific for a particular squamous epithelium. This should be addressed, and the title should be changed to more accurately reflect the findings presented in the manuscript.

2) Further evidence to support co-expression/co-localization of NRG and p63 in SCC would strengthen the manuscript.

3) The use of the blocking antibody is powerful but the details of how specific this reagent is for NRG are lacking. Further evidence of specificity and/or the use of genetics to support the findings are necessary.

4) The authors should clarify whether tumors regress or arrest after treatment with the antibody. A waterfall plot of all of the tumors would be helpful.

5) Is there squamous differentiation in cell lines? This needs to be assessed. If so, it is surprising that this did not affect proliferation.

6) Although several experimental models are used, it would be helpful to perform similar experiments in at least three cell lines or to determine whether there are specific squamous tissues in which these findings are most relevant.

For your information, we have included the reviews below, as you might find them helpful when preparing the revised version of your manuscript.

*Reviewer #1:*

In this manuscript Jackson and co-workers investigated the role of p63-NRG in tumor cell proliferation and tumor maintenance. They found that p63 and NRG expression co-occurred in the TCGA lung and esophageal cancer datasets. They showed that suppressing p63 decreased NRG expression and was localized to the NRG. Using a NRG antibody they showed that there was a modest effect on cell proliferation but a stronger effect in some tumor models. They argue that the effects on tumors is due to SCC differentiation.

Overall these experiments are well described and well controlled. The observations are timely and important and may have therapeutic implications. However, there are a few points that require some additional clarification.

The authors argue that the NRG plays a key role in SCC or those that co-express NRG and p63. However, they do not show this directly. It would be helpful to perform experiments in a panel of cell lines in which these markers are noted so that the point can be made clear. This is particularly important since the authors show that some tumors don't respond to anti-NRG, suggesting that they don't co-express p63 and NRG but this is not shown.

It is also not clear whether the tumors regress or simply arrest. It would be helpful to show a waterfall plot for each tumor.

It is also not clear why squamous differentiation would not affect proliferation. Have the authors formally assessed whether there is evidence of squamous differentiation in the 2D cultures?

*Reviewer #2:*

The authors show that NRG1 is a transcriptional target of p63 and argue that anti-NRG1 could be an effective therapy for a wide variety of squamous cell carcinomas. This data is interesting; however, the authors use a very limited number of cell lines and in vivo models. To strengthen the manuscript, the authors should focus on one type of SCC with a more robust number of cell lines and in vivo models.

*Reviewer #3:*

In this manuscript Hedge et al. show that TP63 directly regulates NRG1 expression in squamous cell carcinoma (SCC) cell lines. In addition, they show that treatment with an anti-NRG1 antibody inhibits the growth of SCC xenograft models and provide evidence that inhibition of NRG1 induces keratinization and terminal squamous differentiation of tumor cells. Although the data are potentially interesting, there are some major issues with the experimental design, which include the following:

1) All experiments aimed at elucidating NRG1 function utilize an NRG1 blocking antibody. However, the authors fail to demonstrate (e.g. by using genetic tools) that the phenotype observed in their model systems upon antibody treatment is specifically mediated by NRG1 inhibition.

2) NRG1 has already been established as a TP63 target in breast cells and NRG1/ERBB signaling has already been reported to play a role in squamous cell carcinoma biology. The authors provide some evidence that NRG1 inhibition decreases SCC cell proliferation and promotes cell differentiation. This hypothesis should be further investigated in properly controlled in 3D cell culture models where the role of ERBB3 and ERBB4 signaling could also be addressed.

3) In Figure 1, it would be important to perform double IHC/IF to demonstrate co-localization of deltaN-TP63 and NRG1 at the protein level in tumor cells within human SCC tissue samples.

4) In Figure 3, immunoblotting data documenting inhibition of *Nrg1/ErbB* signaling in the treated in vivo models should be provided.

5) In Figure 4, data for anti-Ragweed control (3 dose) should be showed for all experiments.

---

## [Author Response]

Essential revisions:1) The title is quite broad. The reviewers were unclear whether the findings are true in all squamous tissues or are specific for a particular squamous epithelium. This should be addressed, and the title should be changed to more accurately reflect the findings presented in the manuscript.

We revised the title to better reflect the findings. The new title is “NRG1 is a Critical Regulator of Differentiation in TP63-driven Squamous Cell Carcinoma”.

2) Further evidence to support co-expression/co-localization of NRG and p63 in SCC would strengthen the manuscript.

This is a great suggestion to demonstrate the co-localization of NRG1/p63. Unfortunately, the available NRG1 antibodies do not work for immunofluorescence, IHC or even western blot. We have tested many different antibodies for NRG1 in the past and have found them to be non-selective. We always perform shRNA knockdown of NRG1 to validate any commercial antibodies and so far, we have not found one that shows a specific staining pattern. Moreover, the blocking antibodies used in this paper are also not suitable for those applications. However, we would like to bring to the reviewer’s attention that we have provided TP63, NRG1, ERBB3 and ERBB4 mRNA levels in a table format just below the in vitro growth assay in Figure 2.

3) The use of the blocking antibody is powerful but the details of how specific this reagent is for NRG are lacking. Further evidence of specificity and/or the use of genetics to support the findings are necessary.

Details on antibody selectivity can be found in our earlier publication (Hegde et al., 2013). We have added a sentence pointing the readers to this publication for additional details on the antibody properties.

4) The authors should clarify whether tumors regress or arrest after treatment with the antibody. A waterfall plot of all of the tumors would be helpful.

We have included a waterfall plot of all tumors as a supplementary figure. The tumors indeed undergo stasis not regression, consistent with the role of NRG1 in regulating differentiation, not cell survival. As discussed in our final paragraph, EGFR and other signaling pathways also play an important role in driving these tumors and inhibition of NRG1 could drive deeper responses when combined with EGFR inhibition.

5) Is there squamous differentiation in cell lines? This needs to be assessed. If so, it is surprising that this did not affect proliferation.

We thank the reviewers for raising this critical question. We carried out qPCR analysis of four differentiation markers in three different squamous cell carcinoma lines treated with anti-NRG1 or control antibody in vitro. The result are provided in Figure 4—figure supplement 3. Consistent with our in vivo findings, the cell lines treated in vitro also showed significant increases in multiple differentiation markers. However, the magnitude of induction was much lower in vitro, consistent with the more modest effects on proliferation observed in vitro compared to the tumor growth inhibition in vivo.

6) Although several experimental models are used, it would be helpful to perform similar experiments in at least three cell lines or to determine whether there are specific squamous tissues in which these findings are most relevant.

We have conducted additional experiments as mentioned above with 3 cell lines; FaDu (Head and Neck), HCC95 (lung) and KYSE-180 (esophageal) squamous cell carcinomas. The induction of differentiation markers upon treatment was highest in the lung and esophageal lines.